# *Trypanosoma brucei rhodesiense* Inhibitor of Cysteine Peptidase (ICP) Is Required for Virulence in Mice and to Attenuate the Inflammatory Response

**DOI:** 10.3390/ijms24010656

**Published:** 2022-12-30

**Authors:** Tatiana F. R. Costa, Amy Goundry, Alexandre Morrot, Dennis J. Grab, Jeremy C. Mottram, Ana Paula C. A. Lima

**Affiliations:** 1Instituto de Biofisica Carlos Chagas Filho, Universidade Federal do Rio de Janeiro, Rio de Janeiro 21941-901, Brazil; 2Laboratório de Imunoparasitologia, Faculdade de Medicina, Universidade Federal do Rio de Janeiro, Rio de Janeiro 21941-900, Brazil; 3Centro de Pesquisa em Tuberculose, Fundação Oswaldo Cruz (FIOCRUZ), Manguinhos 21040-900, Brazil; 4Department of Pathology, Uniformed Services University of the Health Sciences, Bethesda, MD 20814, USA; 5York Biomedical Research Institute and Department of Biology, University of York, York YO10 5DD, UK

**Keywords:** protease, inhibitor, chagasin, inflammation, Trypanosoma, virulence

## Abstract

The protozoan *Trypanosoma brucei rhodesiense* causes Human African Trypanosomiasis, also known as sleeping sickness, and penetrates the central nervous system, leading to meningoencephalitis. The Cathepsin L-like cysteine peptidase of *T. b. rhodesiense* has been implicated in parasite penetration of the blood–brain barrier and its activity is modulated by the chagasin-family endogenous inhibitor of cysteine peptidases (ICP). To investigate the role of ICP in *T. b. rhodesiense* bloodstream form, *ICP*-null (Δ*icp*) mutants were generated, and lines re-expressing *ICP* (Δ*icp:ICP*). Lysates of Δ*icp* displayed increased E-64-sensitive cysteine peptidase activity and the mutant parasites traversed human brain microvascular endothelial cell (HBMEC) monolayers in vitro more efficiently. Δ*icp* induced E-selectin in HBMECs, leading to the adherence of higher numbers of human neutrophils. In C57BL/6 mice, no Δ*icp* parasites could be detected in the blood after 6 days, while mice infected with wild-type (WT) or Δ*icp:ICP* displayed high parasitemia, peaking at day 12. In mice infected with Δ*icp*, there was increased recruitment of monocytes to the site of inoculation and higher levels of IFN-γ in the spleen. At day 14, mice infected with Δ*icp* exhibited higher preservation of the CD4^+^, CD8^+^, and CD19^+^ populations in the spleen, accompanied by sustained high IFN-γ, while NK1.1^+^ populations receded nearly to the levels of uninfected controls. We propose that ICP helps to downregulate inflammatory responses that contribute to the control of infection.

## 1. Introduction

The protozoan parasite *Trypanosoma brucei rhodesiense* is the pathogen responsible for human sleeping sickness, a fatal disease that causes meningoencephalitis [1]. The parasite is transmitted by the bite of infected *Glossina* spp. flies and *T. b. rhodesiense* bloodstream forms (BSF) replicate in the blood and lymph, while parasite reservoirs have been described in the adipose tissue [2] and skin [3]. Antibody-mediated immunity partially controls parasite burden, typically via response to the main variant surface glycoprotein (VSG), but a sophisticated mechanism of antigenic variation via gene switch enables the parasite to escape, provoking repeated waves of blood parasitemia [4]. Meningoencephalitis takes place after penetration of the parasite in the central nervous system, by mechanisms that are not yet fully understood. *T. b. brucei*, a subspecies that is not infective to humans, is often used as an experimental model and has been extensively characterized at the molecular and biochemical levels [5,6]. *T. b. brucei* is susceptible to lysis by innate immune complexes naturally present in human serum, named trypanolytic lysis factor 1 and 2 (TLF), while *T. b. rhodesiense* is refractory to lysis by TLFs [7,8].

Parasite proteases belonging to the C1A papain-like family cysteine peptidases similar to cathepsin L and cathepsin B were long associated with parasite virulence in vitro and in vivo [9]. The lysosomal cathepsin L-like protease (TbCATL), also named rhodesain, brucipain or trypanopain, is encoded by a multi-gene family, and application of the interference RNA (RNAi) technique suggested that the enzyme plays a role in parasite virulence to mice [10]. Furthermore, the use of irreversible inhibitors to TbCATL in a model of the blood–brain barrier (BBB) in vitro, using a recognized human brain microvascular endothelial cell line [11,12,13], provided evidence that the enzyme contributes to the traversal of BBB by *T. b. rhodesiense* [14]. TbCATL was also implicated in cardiac disfunction in a rat model of infection, via increases in the arrhythmogenic spontaneous diastolic sarcoplasmic reticulum (SR)-mediated Ca^2+^ release in cardiomyocytes [15]. Although there has not yet been genetic validation for TbCATL essentiality, the use of different peptidyl inhibitors to papain-like peptidases has provided chemical evidence that TbCATL is crucial for parasite survival [16]. The potential of TbCATL as a drug target has driven an active search for inhibitors of different chemical entities over the years [17,18,19,20]. More recently, the use of a selective inhibitor to TbCATL indicated that the enzyme is essential for parasite survival [21].

*Trypanosoma brucei* spp. express an endogenous inhibitor of approximately13 kDa named Inhibitor of Cysteine Peptidase (ICP) [22]. ICP is part of a large family of inhibitors of papain-like cysteine proteases, the I42 chagasin-family, with members distributed among protozoa, bacteria, and archaea [23,24]. ICP has high affinity for TbCATL and there is genetic evidence that *T. b. brucei* ICP modulates the activity of TbCATL in vivo [25]. *ICP*-null mutants in *T. b. brucei* show increased degradation of endocytosed VSG, helping immune evasion, and exhibited higher blood parasitemia in BALB/c mice [25]. Additional phenotypes included improved differentiation in vitro from the bloodstream to the procyclic cell form (life stage found in the insect vector) accompanied by accelerated kinetics of exchange of cell surface VSG for the procyclic protein, procyclin. Later on, a genome-wide screen to search for factors that render *T. b. brucei* sensitive to lysis by human serum identified ICP as a pivotal factor controlling sensitivity to lysis via the modulation of TbCATL [26]. *ICP*-null mutants in *T. b. brucei* were 7-fold less susceptible to lysis by normal human serum and this phenotype was mediated by increased TbCATL activity, denoting that the balance between the TbCATL/ICP levels might be crucial for the survival of *T. b. brucei* [26]. Here, we have addressed the role of ICP in *T. b. rhodesiense* by the generation of *ICP*-null mutants and found that the phenotype was different from those displayed by *T. b. brucei*. We describe that *T. b. rhodesiense* ICP is required for parasite survival in mice and is implicated in the attenuation of the inflammatory immune response of the host.

## 2. Results

*ICP*-null mutants (Δ*icp*) were generated by homologous recombination in *T. b. rhodesiense* bloodstream form (BSF), via replacement of both alleles of the *ICP* gene with cassettes containing the genes for antibiotic resistance to blasticidin and hygromycin, respectively (Figure 1A). The replacement of *ICP* by the recombination cassettes in parasite clones resistant to both antibiotics was verified by PCR and subsequently, a complemented line (Δ*icp*:*ICP*) was generated by introducing a copy of *ICP* into the tubulin locus via homologous recombination (Figure 1B). Southern blotting confirmed the genotype of Δ*icp* and Δ*icp*:*ICP* lines (Figure 1C,D).

To check if ICP levels contribute to the modulation of cysteine peptidase activity in *T. b. rhodesiense*, we measured peptidolytic activity in parasite lysates using a fluorogenic substrate (Figure 2). In BSF lysates derived from Δ*icp*, there was a 50% increase in the peptidolytic activity, as compared to lysates from wild-type (WT) or the complemented line, denoting increased cysteine peptidase activity in the absence of ICP. In BSF lysates from all tested lines, there was a prominent reduction in peptidolytic activity upon incubation with the wide-range irreversible inhibitor of papain-like enzymes, E-64, and a modest decrease with CA-074, an inhibitor of cathepsin B, indicating that the activities from papain-like peptidases were preferentially detected in the assay.

BSF Δ*icp* displayed slight, but significant, reduced growth in vitro in comparison to WT, but this phenotype was not reverted in the complemented line, Δ*icp*:*ICP* (Figure 3). All attempts to generate anti-serum against recombinant ICP failed, precluding the verification of the levels of ICP protein in the parasite lysates.

We had previously demonstrated that TbCATL contributes to the ability of *T. b. rhodesiense* to induce transient increases in the permeability of human brain microvascular endothelial cell (HBMEC) monolayers in vitro by decreasing the trans-endothelial electric resistance, and that this is associated with parasite capacity to transmigrate through HBMEC monolayers (an experimental model for the BBB) [14,27]. We next asked if the absence of ICP would affect parasite traversal of the BBB model. Throughout the kinetics, Δ*icp* BSF traversed the HBMEC monolayers nearly 2-fold more efficiently than WT and Δ*icp:ICP*, indicating that the lack of ICP improves the parasite’s capacity to penetrate the BBB, and this was fully restored to WT levels in the Δ*icp:ICP* line (Figure 4).

Next, we asked if *T. b. rhodesiense* could induce the activation of HBMECs, and if ICP had any influence in this process. To test that, HBMEC monolayers were incubated with parasites and the surface expression of the activation marker E-selectin (CD62E) was determined. Cell surface E-selectin was promptly detected in cells that had been exposed to parasites, as compared to control cells incubated with medium alone (Figure 5A). HBMECs exposed to Δ*icp* displayed nearly 3-fold higher amounts of surface E-selectin, as compared to those exposed to WT or Δ*icp:ICP* parasites, associating the lack of ICP with a higher potential to induce endothelial activation. To address whether the changes induced by *T. b. rhodesiense* on the endothelial cells affected their interaction with leukocytes, the capacity of HBMECs to retain resting human neutrophils attached to their surface was used as a parameter. Twice as many neutrophils were found adhered to HBMECs that had been previously exposed to WT *T. b. rhodesiense* as compared to control HBMECs (Figure 5B), while there were 3-fold higher numbers of neutrophils adhered to HBMECs that had been previously exposed to Δ*icp*. We evaluated further whether the increased neutrophil adhesion was promoted by E-selectin by using neutralizing antibodies (Figure 5C). HBMECs that had been exposed to WT or Δ*icp* were treated with anti-E-selectin, prior to incubation with neutrophils. Treatment with antibodies to E-selectin prevented the increased adhesion of neutrophils promoted by HBMECs pre-exposed to Δ*icp*, bringing the number of adhered neutrophils to the level observed in WT-exposed HBMECs. These observations suggest that endothelial activation directly by *T. b. rhodesiense* has the potential to modify the microvasculature-immune system interface and that ICP plays a role in the parasite’s ability to induce endothelial inflammation.

To test whether endothelial activation by the parasite also influenced its interaction with human T lymphocytes, we measured the adherence of CD4^+^ and CD8^+^ T cells to HBMEC monolayers (Figure 6). First, lymphocytes directly isolated from peripheral blood were tested, and approximately 50% of HBMECs were found to be capable of retaining CD4^+^ lymphocytes (Figure 6A), or CD8^+^ lymphocytes (Figure 6B), in control conditions. HBMECs that had been previously exposed to *T. b. rhodesiense* WT or to Δ*icp:ICP* displayed significantly increased T cell adherence. In contrast, HBMECs that had been previously exposed to Δ*icp* did not display increased capacity to retain CD4^+^ or CD8^+^ T lymphocytes as compared to controls or to HBMECs previously exposed to WT or to Δ*icp:ICP* (Figure 6A,B). This indicates that ICP affects endothelial cells in opposite ways regarding their capability to interact with leukocytes from the innate versus the adaptive immune system. We further tested how endothelial activation conveyed by the parasite affected the interaction with activated lymphocytes. To that end, we incubated purified T lymphocytes with phytohemagglutinin in vitro, prior to performing co-culture with HBMECs. Pre-exposure of HBMECs to WT *T. b. rhodesiense* maximized T cell retention to nearly 80%, revealing that the parasite can directly alter the status of microvascular cells to enhance its potential to retain activated CD4^+^ T cells (Figure 6C,D). Parasites lacking ICP not only lost this potential but also made HBMECs less capable to retain activated CD4^+^ T cells (Figure 6C). A similar pattern was observed when the adhesion of CD8^+^ T cells were analyzed (Figure 6D). In all cases, the complemented parasite line Δ*icp:ICP* behaved similarly to WT parasites, supporting a role for ICP in the modulation of endothelial cell properties affecting the interaction with T lymphocytes.

Given the observations that ICP plays a role in the ability of the parasite to alter the properties of endothelial cells and how they interact with leukocytes, we adopted the mouse model to address whether ICP could further influence the host–parasite interaction. We had previously described an infection model for *T. b. rhodesiense* using C57BL/6 mice, that displays sequential waves of blood parasitemia and visible signs of motor disability [28]. In the present study, infected mice displayed the first wave of parasitemia around day 4 post-infection and the parasite loads in mice infected with WT parasites were approximately 10-fold higher than those observed in mice infected with Δ*icp* (Figure 7). Mice infected with the complemented line Δ*icp:ICP* also displayed reduced parasitemia during the first wave. Since there should not be antibody-mediated anti-parasite immunity at day 3–4, the low parasite burdens in the first wave could be a reflection of the slower growth observed in vitro for Δ*icp* and Δ*icp:ICP*. At day 10, a robust parasitemia wave commenced, peaking at day 12–13 in mice infected with WT or Δ*icp:ICP* parasites, while we could not detect parasites in the blood of mice infected with Δ*icp*. These results indicate that ICP is essential for maintaining an infection in mice over 14 days.

Given that we observed a higher inflammatory potential of Δ*icp* towards human endothelial cells with consequences to leukocyte adhesion, and that the mice were able to control parasitemia at day 10, when adaptive immunity is building, we evaluated the immune response of infected mice during the early (day 2) and later (day 14) infection stages. First, we analyzed the cell populations present at day 2 at the site of parasite inoculation, the peritoneal cavity. The population profile would indicate if ICP influenced the early innate responses that determine cellular recruitment. We found no alterations in the frequencies of CD11c^+^ dendritic cells (DCs) (Figure 8A) and in the total myeloid CD11b^+^ population (Figure 8B) among infected and uninfected mice. We further dissected the cellular subsets and observed no alterations in the resident F4/80^+^ macrophages among those mice (Figure 8C). In contrast, there was a 2-fold increase in the percentage of Ly6C^+^ cells in the peritoneal cavity of mice infected with WT or with Δ*icp:ICP* parasites, as compared to uninfected mice, denoting active recruitment of monocytes to the infection site (Figure 8D). Furthermore, mice infected Δ*icp* had a significant increase in the percentage of Ly6C^+^ cells in the peritoneal cavity as compared to other infected mice, suggesting that in the absence of ICP, there is increased innate inflammatory responses at tissue sites directly exposed to the parasite. This pattern was likewise observed within the Ly6C^+^/Ly6G^+^ sub-population, supporting increased cellular recruitment to infected tissues in the absence of ICP (Figure 8E). Lastly, we analyzed the frequencies of natural killer (NK) cells, as a representative of the lymphocyte subset involved in innate responses that can be recruited to tissues upon injury and/or infection. While there were decreases in the numbers of NK cells in the peritoneal cavity of mice infected with WT or with Δ*icp:ICP* parasites, as compared to uninfected mice injected with vehicle alone, mice infected with Δ*icp* parasites sustained the increased levels of NK cells (Figure 8F). These results show that in mice infected with Δ*icp*, higher levels of innate cells are maintained at the site of infection in the early stage, which could presumably help to contain infection.

The spleen cellular population was then analyzed at day 2 to verify if there were detectable alterations during initial infection. As anticipated, we did not observe alterations in the lymphocyte subsets of adaptive response, CD4^+^ T cells (Figure 9A), CD8^+^ T cells (Figure 9B), CD19^+^ B cells (Figure 9C), and neither in the NK subset (Figure 9D). Monocytes (Figure 9E) and neutrophils (Figure 9F) were likewise unaltered in infected mice as compared to control mice, while there were noticeably less DCs (Figure 9G) and resident macrophages (Figure 9H) in infected mice. We then selected IFN-γ and TNF-α, which are hallmarks of inflammation, to address whether there were any functional alterations in the spleen environment at day 2. We detected a substantially significant production of IFN-γ by splenocytes derived from mice infected with Δ*icp*, as compared to those infected with WT or Δ*icp*:*ICP* parasites and control mice (Figure 9I), denoting increased inflammation during early infection. The production of TNF-α, albeit also increased, was too variable to allow definite conclusions (Figure 9J).

Considering that mice infected with Δ*icp* lacked detectable blood parasitemia at day 13, we verified the status of the immune cells in the spleen at day 14. As expected, the cellularity was overall higher in the spleens of infected mice as compared to uninfected controls (Figure 10A). However, there was a significant decrease in the cellularity of mice infected with Δ*icp*, which could reflect that the immune response was receding. However, the analyses of lymphocyte subsets showed that mice infected with Δ*icp* retained a significantly higher number of T and B cell subsets in the spleen (Figure 10B–D), as compared to mice infected with WT or Δ*icp*:*ICP* parasites, which could in principle, contribute to specific anti-parasite immunity. In contrast, cellular subsets typically related to innate responses, such as NK cells (Figure 10E), myeloid cells (Figure 10F), macrophages (Figure 10G), and neutrophils (Figure 10H), were decreased in the spleens of mice infected with Δ*icp,* as compared to other infected mice, while still sustained at higher levels as compared to uninfected controls. The cytokine profile showed an equally high inflammatory response in all infected mice, as determined by levels of IFN-γ (Figure 10I), TNF-α (Figure 10J), and IL-6 (Figure 10K). Taken together, these results indicate that in the absence of ICP, the host not only triggers an inflammatory response earlier, but also sustains it throughout infection, while preserving the lymphocyte subset that could ultimately help in the control of parasite burden.

## 3. Discussion

The lysosomal cysteine peptidase of *T. brucei*, TbCATL, has been extensively implicated in parasite virulence, and we and others have provided evidence that its activity is finely tuned by its endogenous inhibitor ICP, at least so in the *T. b. brucei* subspecies [22,23]. We set out to address the role of ICP in the human-infective *T. b. rhodesiense* subspecies, as a model with potential relevance for pathogenicity. This is especially pertinent given that we have shown that this parasite uses TbCATL to induce transient increases in endothelial permeability in a model of HBMECs, with potential consequences to the traversal of the BBB [14]. We have also previously identified protease-activated receptor 2 (PAR2) as among the targets for TbCATL in HBMECs, through which increased intracellular Ca^2+^ signals are conveyed, resulting in increases in vascular permeability [29]. We postulated that if endogenous ICP controls TbCATL, it could play a pivotal role in the way *T. b. rhodesiense* interacts with HBMECs. We were able to successfully generate *ICP*-null mutants in BSF *T. b. rhodesiense* and detected higher levels of cysteine peptidase activity in their parasite lysates. Even though the Z-FR-MCA substrate used in the assays is not selective to papain-like cysteine peptidases, the use of the selective inhibitor E-64 allowed us to conclude that the great majority of the activity detected was that of papain-like peptidases. *T. brucei* has two major papain-like enzymes, TbCATL and TbCATB, which could be targets for ICP in the parasite. Although the inactivation constant of ICP to TbCATB was not determined, work done with the *T. cruzi* orthologue chagasin, showed that the Ki values for human CATB are 10-fold higher than for human CATL and for the *T. cruzi* CATL (cruzipain). It is possible that ICP behaves similarly, and displays higher affinity for TbCATL than for TbCATB, albeit being an effective inhibitor of both enzymes. Furthermore, we cannot rule out that additional parasite cysteine peptidases are also subject to inhibition by ICP, particularly those peptidases belonging to the C2 family. TbCATL has been previously implicated in parasite growth in vitro, therefore, we did not expect and that increased TbCATL activity in *ICP*-null mutants would be paralleled by diminished parasite multiplication. However, this phenotype was not complemented in the Δ*icp*:*ICP* line, suggesting that it might be a bystander effect due to genetic manipulation, and unrelated to ICP or TbCATL. Alternatively, it is possible that levels of ICP expression in the complemented line did not match those of wild-type parasites, preventing full complementation. We have not addressed if deletion of *ICP* affected the differentiation of “long slender” multiplicative forms to the “short stumpy” forms, which could alternatively explain the difference in parasite multiplication of Δ*icp*. Since we have been unable to determine the subcellular localization of ICP to date, we cannot rule out that ICP and TbCATL may not reside entirely in the same subcellular compartment, making the control of the enzyme by ICP a regulated event that is not fully restored in the complemented line. On the other hand, we observed that the Δ*icp*:*ICP* line exhibited next to full reversal of all phenotypes displayed by Δ*icp* when addressing the parasite–HBMEC interaction.

In agreement with a role for TbCATL in BBB penetration, Δ*icp* parasites traversed HBMEC monolayers more efficiently, which could be linked to increased TbCATL activity. This strengthens our observations that *T. b. rhodesiense* treated with the irreversible inhibitor of TbCATL, K11777, although still viable, was uncapable to induce Ca^2+^ signalling on HBMECs and to traverse the BBB [14]. Of relevance to neuroinflammation, we found that *T. b. rhodesiense* directly induces the expression of E-selection at the cell surface of HBMECs, even in the absence of other external stimuli. Furthermore, this translated into increased capacity to bind and retain human neutrophils and activated T lymphocytes. The endothelial–leukocyte interaction is among the hallmarks of inflammation and the extent of adhesion contributes to leukocyte transmigration into tissues [30]. Therefore, the sites for transmigration are largely controlled by endothelial properties to protect sensitive tissues, such as the eye and the brain, from excessive cellular infiltration and potential tissue damage [31]. More specifically, vessels with BBB properties have reduced trafficking of leukocytes due to low expression of adhesion molecules [30]. Brain endothelial cells, contrary to most endothelial cells, do not contain P-selectin in their Weibel–Palade bodies [32], and express at least 10-fold less E-selectin in response to inflammation as compared to endothelium of other organs [33]. We found that the induction of E-selectin in HBMECs by the parasite is negatively controlled by ICP, which in turn, reduces the capacity of the brain endothelium cells to perform lasting interaction with neutrophils. If we assume that increased induction of E-selectin is a consequence of increased TbCATL activity, we may conclude that this parasite peptidase plays a crucial role in the extent of BBB activation, enhancing the chances for barrier disruption and neuroinflammation, in addition to transient permeability. Our own microarray data of HBMECs exposed to *T. b. rhodesiense* treated with TbCATL inhibitor showed that enzyme inhibition led to increased expression of several transcription factors such as GATA3 and HMGB1 [29], exemplifying how, by analogy, the levels of ICP controlling TbCATL can have consequences to how the brain microvasculature responds to the parasite. We cannot rule out that ICP can also control the activity of cysteine peptidases of the host, helping to keep proteolysis controlled under inflammatory settings.

Surprisingly, we observed an opposite effect with respect to T cell adhesion, i.e., HBMECs exposed to Δ*icp* were more refractory to adhesion by T lymphocytes. Although we have not addressed the specific adhesion molecules downmodulated in those interactions, it is possible that excessive TbCATL activity could lead to the cleavage of surface molecules of HBMECs, diminishing lymphocyte adhesion. In experimental mouse infections, CD4^+^ T lymphocytes are known to penetrate the CNS in an IFN-γ-dependent way, helping in tissue damage [34]. Therefore, the TbCATL/ICP balance could further influence the extent of CD4^+^ T cells interaction with the brain endothelium, with consequences to pathology.

Finally, we addressed the influence of ICP on parasite fitness in the mouse model. The infection with Δ*icp* was controlled by the mouse, leading to undetectable blood parasitemia at day 10, which supports that ICP is virulence factor for *T. b. rhodesiense*. This is in contrast to the phenotype we described for *T. b. brucei ICP*-null mutants, which were more infective to BALB/c mice [25], revealing fundamental differences in how those two subspecies interact with the mammalian host. The monomorphic *T. b. brucei* 427 strain used in our previous study multiplies only as “longer slender” forms which undergoes exponential growth, killing BALB/c mice in a few days. We discarded the possibility that the discrepant phenotypes between *T. b. brucei* Δ*icp* and *T. b. rhodesiense* Δ*icp* resulted from differences in the susceptibility of BALB/c versus C57BL/6 because we observed that BALB/c mice also controlled blood parasitemia of *T. b. rhodesiense* Δ*icp*, and survived longer (unpublished data). Although we have not directly addressed the mechanisms involved in parasite killing, it is well established that parasite phagocytosis by macrophages and monocytes is crucial for parasite elimination [4], which is optimized by parasite opsonization mainly via anti-VSG antibodies [35]. The initial parasitemia peak was similarly lower in mice infected with Δ*icp* or Δ*icp*:*ICP,* as compared to those of mice infected with WT parasites. However, Δ*icp*:*ICP* gave rise to a second parasitemia wave similar to that of WT parasites peaking at day 13, while we could no longer detect Δ*icp*. At day 2 post-infection, the levels of splenic IFN-γ were increased in mice infected with Δ*icp*, which could presumably enhance macrophage/monocyte activation early on and help to control parasite burden. Additionally, there were increased levels of Ly6C^+^ cells in the peritoneal cavity of mice infected with Δ*icp*, which is a sign of increased inflammation. Levels of IL12 and IL10 were similar among all mice groups both at day 2 and at day 14 (data not shown). Although we have not determined the levels of inflammatory cytokines in the peritoneal exudate, we speculate that if levels of IFN-γ and TNF-α, for example, are also elevated at the inoculation site at day 2, it could help to activate myeloid cells to become more trypanocidal.

At day 14, mice infected with Δ*icp* showed significant preservation of the lymphocyte pool, which can be crucial for anti-parasite immunity. Indeed, it is known that in mice infected with *T. b. brucei*, the initial peak of parasitemia occurs independently of B cells, while those cells are crucial to control following parasitemia peaks [35]. Nevertheless, after the second wave, there is a rapid reduction in splenic marginal zone and follicular B cells, as well as CD8^+^ T cells and both NK and natural killer T (NKT) cells [36], which can contribute to parasite immune evasion. Furthermore, NK cells have been described as the main contributors to B cell depletion in the spleen in experimental infections with *T. brucei* [37]. It is noteworthy that the percentage of splenic NK cells was markedly reduced at day 14 in mice infected with Δ*icp*, which can offer an explanation to why those mice were able to maintain a 3-fold higher percentage of B cells. Finally, the combination of higher sustained IFN-γ levels exerting a microbicidal role for macrophages, combined with a greater availability of T and B cells, could act in conjunction to control parasite burden.

In summary, we conclude that ICP plays a pivotal role in *T. b. rhodesiense*, allowing the parasite to suppress inflammatory responses of the host at several levels, such as vasculature activation, myeloid cell recruitment, and the production of inflammatory cytokines with consequences to parasite fitness and survival.

## 4. Material and Methods

### 4.1. Ethics Statement

All animal procedures were undertaken in adherence to experimental guidelines and procedures approved by the Ethics Committee on Animal Use of UFRJ (Comissão de Ética no Uso de Animais, CEUA) 03/15–UFRJ. All experiments were performed in accordance with the guidelines and regulations of the National Council of the Control of Animal Experimentation (Conselho Nacional de Controle de Experimentação Animal, CONCEA)—Resolução Normativa No. 39, Brasília, 20 June 2018.

### 4.2. Parasites

Bloodstream form (BSF) parasites of *T. b. rhodesiense* IL1852 [14] were grown in HMI-9 medium (Gibco) supplemented with 10% fetal bovine serum (FBS; Gibco) and 10% serum plus (JRH Biosciences), and incubated at 37 °C and 5% CO_2_. Parasites were passed to fresh medium every 2 days. Antibiotic concentrations used for selection were 5 μg/mL hygromycin B (Calbiochem, La Jolla, CA, USA), 10 μg/mL blasticidin (Calbiochem) and 1.5 μg/mL phleomycin (InvivoGen, San Diego, CA, USA). For growth curves, parasites were seeded in medium at a concentration of 10^4^ parasites/mL and counted daily using a Neubauer chamber on an inverted light microscope. On the third and fifth days, parasites were diluted to the initial concentration and counting continued. The cumulative cell number was calculated by multiplying by the dilution factor.

### 4.3. Generation of Transgenic Lines

Parasite lines deficient in *ICP* (Δ*icp*) and re-expressing *ICP* (Δ*icp:ICP*) were generated by the sequential gene replacement of both alleles of the wild-type (WT) gene locus and the subsequent re-introduction of the gene into the tubulin locus. Genomic DNA from *T. b. rhodesiense* 1852 WT parasites was extracted using the DNeasy Blood & Tissue kit (Qiagen). To generate the *ICP* knock-out constructs, the 5′ and 3′ *ICP* flanking regions were amplified from the genomic DNA by PCR using Taq DNA polymerase with primers OL1609/OL1610 for 5′*ICP* and OL1611/OL1612 for 3′*ICP* (Table 1). The resulting PCR products were individually sub-cloned into the pGEM-T-Easy vector (Promega), which was confirmed by sequencing. The 5′and 3′ *ICP* flanking regions were digested from these vectors with *Not*I/*Xba*I and *Apa*I, respectively, and ligated into similarly digested plasmids with blasticidin and hygromycin resistance cassettes to generate the plasmids pGL1149 and pGL1151, respectively. To generate the *ICP* re-expression construct, the *ICP* gene was amplified from the genomic DNA with primers NT90/NT91 [22] using Pfu Turbo polymerase and cloned into the pPCR Script vector (Stratagene). After confirming the sequence, the gene was excised from the sub-cloning vector using *Xho*I/*Xba*I and ligated into a similarly digested plasmid containing a phleomycin resistance cassette to generate the plasmid pGL1493. The plasmids, pGL1149 and pGL1151, were digested with *Not*I, and the linearized DNA was gel extracted (Qiagen, Hilden, Germany) and subsequently ethanol precipitated. In total, 10^7^ mid-log parasites were electroporated in 100 µL Human T Cell Nucleofector solution (Lonza) with 20 µg DNA using an Amaxa Nucleofector. Parasites were cloned in 24-well plates in the presence of the selection antibiotics. Deletion and integration in the correct genome location of selected clones was confirmed by PCR and Southern blotting.

### 4.4. Southern Blotting

For each cell line generated, approximately 5 μg of genomic DNA was digested overnight with the *Stu*I and *Sph*I and separated by electrophoresis in 0.8% agarose gels. The gel was washed for 10 min in 0.25 M HCl, rinsed in distilled water then washed with denaturation buffer (1.5 M NaCl, 0.5 M NaOH) for 15 to 30 min before being rinsed again with distilled water. Finally, the gel was washed with neutralization buffer (3 M NaCl, 0.5 M Tris-HCl, pH 7.0) for 30 min and rinsed with distilled water. DNA was transferred overnight to a Hybond N nylon membrane (GE Healthcare, Little Chalfont, UK) by capillary force in 20× SSC buffer (3 M NaCl, 0.3 M sodium citrate, pH 7.0). After the transfer, the membrane was washed for 10 min in 2× SSC buffer and the DNA was crosslinked to the membrane in a UV Stratalinker 2400 crosslinker (Stratagene, La Jolla, CA, USA) at 1200 mJ. For the probing, the Gene Images Alk-Phos Direct Labelling and Detection System (GE Healthcare Little Chalfont, UK) was used, according to the manual, to generate a fluorescent-labeled DNA probe. The 5′ flanking region of *ICP* or the open reading frame (ORF) of *ICP*, digested out of the respective plasmid and purified from agarose gel, were used as probes. The signal was detected using CDP-Star detection reagent (GE Healthcare).

### 4.5. Cysteine Peptidase Activity Assays

BSF parasites (10^6^) were lysed in 1% NP-40 in PBS for 30 min. Peptidase activity was tested in 0.1 M Na_2_HPO_4_, 0.2 M NaCl, 5 mM EDTA pH 6.5, 2.5 mM DTT using 3 µM of Z-Phe-Arg-MCA (Sigma, St. Louis, MO, USA) as a substrate. To distinguish between the activity of cathepsin B (TbCATB) and TbCATL, 10 µM of the selective cathepsin B inhibitor, CA-074, or the wide-range inhibitor of papain-like enzymes, E-64 (Sigma), were used, respectively, and were added to the lysates 10 min prior to the addition of the substrate.

### 4.6. Transmigration Assays

The immortalized human brain microvascular endothelial cell line (HBMEC) for in vitro human BBB tightness and paracellular permeability studies was previously described elsewhere [11,12,13,14]. The immortalized cells form polarized, cobblestone monolayers that express a clear human brain microvascular endothelial phenotype [11,27,38,39]. HBMECs were cultivated in M199 medium supplemented with 10% FBS and 5 × 10^3^ cells were seeded on top of Transwell ^TM^ inserts (Costar; Corning Inc.) containing 3 μm pores. Cells were cultured in the inserts at 37 °C and 5% CO_2_ for approximately 5 days. BSF parasites (10^6^) were collected by centrifugation at 1500× *g* for 10 min, resuspended in pre-warmed 50% M199, 50% HMI-9, 20% FBS. Before addition of the parasites, the medium in the wells containing inserts was changed to 50% M199, 50% HMI-9, 20% FBS and then BSF were added to the top of the HBMEC-containing inserts. The cultures were incubated at 37 °C and 5% CO_2_ for 3, 4, and 5 h, and the number of parasites present in the bottom chamber was determined by counting aliquots on the Neubauer chamber.

### 4.7. Adhesion Assays

HBMECs (5 × 10^4^) were seeded on top of glass coverslips in 24-well culture plates in M199 supplemented with 10% FBS and incubated at 37 °C and 5% CO_2_. After 3 days, the cultures were washed and BSF trypanosomes (10^6^) were added at a 2:1 parasite:cell ratio for 18 h in 50% M199, 50% HMI-9, 20% FBS at 37 °C and 5% CO_2_. Following this, the cells were washed to remove free parasites prior to 1h incubation, at 37 °C, with 10^6^ neutrophils, or for 2 h with 4 × 10^5^ magnetic bead-purified CD4^+^ and CD8^+^ T cells. Neutrophils were purified from human blood using Lymphoprep (Stemcell Technologies, Fisher Scientific, Seattle, WA, USA) and CD4^+^ and CD8^+^ T cells were purified from human blood using magnetic beads (MACS^®^ Technology, Milteny Biotceh, San Diego, CA, USA). Where indicated in the figure legend, T cells were activated (24 h) or not with 5 µg/mL phytohemagglutinin (PHA; Gibco, Grand Island, NY, USA). After the respective incubation periods, the cells were subsequently washed, fixed with 70% methanol, and Giemsa stained. The number of adhered neutrophils were determined by the counting of 100 HBMECs per coverslip under a light microscope. For neutralization experiments, HBMECs were washed, incubated with 10 μg/mL of anti E-selectin antibodies (BD Biosciences) or control IgG for 1 h in complete medium, at 37 °C and 5% CO_2_, prior to washing and co-cultivation with neutrophils.

### 4.8. Mice

Female or maleC57BL/6J mice between 10 and 18 weeks were obtained from the in-house breeding facility at UFRJ. Mice were infected intraperitoneally with 10^5^
*T*. *b. rhodesiense* BSF in 100 μL RPMI. Parasitemia was determined through a minimal incision at the tip of the tail and manual pressure until a drop of blood appeared and 5 μL could be taken and passed to a slide. A coverslip was added onto the blood drop and the parasites were counted on a light microscope and parasitemia was calculated according to the Brener protocol. Complying with principles of animal welfare, all mice were euthanized at day 14.

For the collection of peritoneal cells, mice were euthanized and the peritoneal cavity was washed with ice-cold RPMI, which was subsequently centrifuged at 1500 rpm for 5 min.

### 4.9. Tissue Processing

On the post-infection day indicated in the figure legend, mice were euthanized by CO_2_ inhalation, the spleens were removed and the masses were recorded. Spleens were macerated through nylon in 800 µL RPMI containing a protease inhibitor cocktail (Sigma-Aldrich, St. Louis, MO, USA). The cell suspensions were centrifuged at 1500 rpm for 5 min and the spleen supernatants were collected for cytokine analysis. The cell pellets were resuspended in ACK for 1 min [0.15 M NH_4_Cl, 1 M KHCO_3_, 0.1 M EDTA, pH 7.2] then 5 mL RPMI was added. The cells were centrifuged at 1500 rpm for 5 min, washed three times in RPMI then counted to obtain the total number of cells. Splenocytes (5 × 10^6^) were plated in RPMI supplemented with 10% FBS in 48-well culture plates and incubated at 37 °C and 5% CO_2_ for 24 h, after which the plates were centrifuged at 1500 rpm for 5 min and the supernatants were collected for cytokine analysis. The remaining spleen cells were then stained and assessed by flow cytometry.

### 4.10. Flow Cytometry

For cytometric analysis of HBMECs, cells were plated in a 24-well tissue culture plate at a density of 5 × 10^4^ cells/mL. After 3 days, the cells were washed and incubated with BSF *T. b. rhodesiense* for 18 h at 37 °C. The detection of E-selectin was performed using anti-E-selectin antibody (anti-CD62E, BD Biosciences, San Diego, CA, USA), followed by washing and incubation with a Alexa 488-conjugated anti- mouse antibody. All steps were performed at 4 °C and all centrifugation steps were performed at 1500 rpm for 5 min. Single-cell suspensions from the peritoneal cavity or from the spleen at 6 × 10^6^ cells were washed in PBS and incubated with an anti-Fc-γ III/II (CD16/32) receptor antibody (BioLegend, San Diego, CA, USA) for 10 min. Cells were washed again in PBS then stained with the fluorochrome-conjugated antibodies for 30 min. The following anti-mouse antibodies were used: FITC-CD4 (eBioscience), PE-Cy5-CD8a (eBioscience, ThermoFischer, Whaltam, MA, USA), APC-CD19 (eBioscience), FITC-NK1.1 (BioLegend), FITC- or PE-CD11b (eBioscience),-PE-Cy5-F4/80 (BioLegend), FITC or PE-CD11c (eBioscience), PE-Cy7-Ly6C (BioLegend), APC-Ly6G (eBioscience), and APC-Gr1 (eBioscience). Cells were washed in PBS then resuspended in 100 µL IC Fixation Buffer (eBioscience), incubated for 30 min. All groups were washed and resuspended in 300 µL PBS and at least 10,000 events were acquired on a BD FACSCalibur. Data were analyzed on FlowJo v10. Gating strategies are described in figure legends and the dot plots are given as Appendix A.

### 4.11. ELISAs

The cytokine concentrations of IL-6, IL-10, TNF-α, and IFN-γ in supernatants from splenocytes cultivated for 24 h were evaluated using Duoset kits specific for mouse cytokines from R&D Systems following the manufacturer’s instructions. Plates were read on an Asys Expert Plus microplate reader.

### 4.12. Statistical Analysis

Graphs were generated in the GraphPad Prism 5 software (GraphPad Software Inc, San Diego, CA, USA) and the distribution of data was checked using a Kolmogorov–Smirnoff test and variance was determined by the F-value. Outliers were identified and removed using the Outlier calculator (https://www.graphpad.com/quickcalcs/grubbs1/, accessed in 17 November 2022). The appropriate statistical tests were then applied as described in the figure legends.

## Figures and Tables

**Figure 1 ijms-24-00656-f001:**
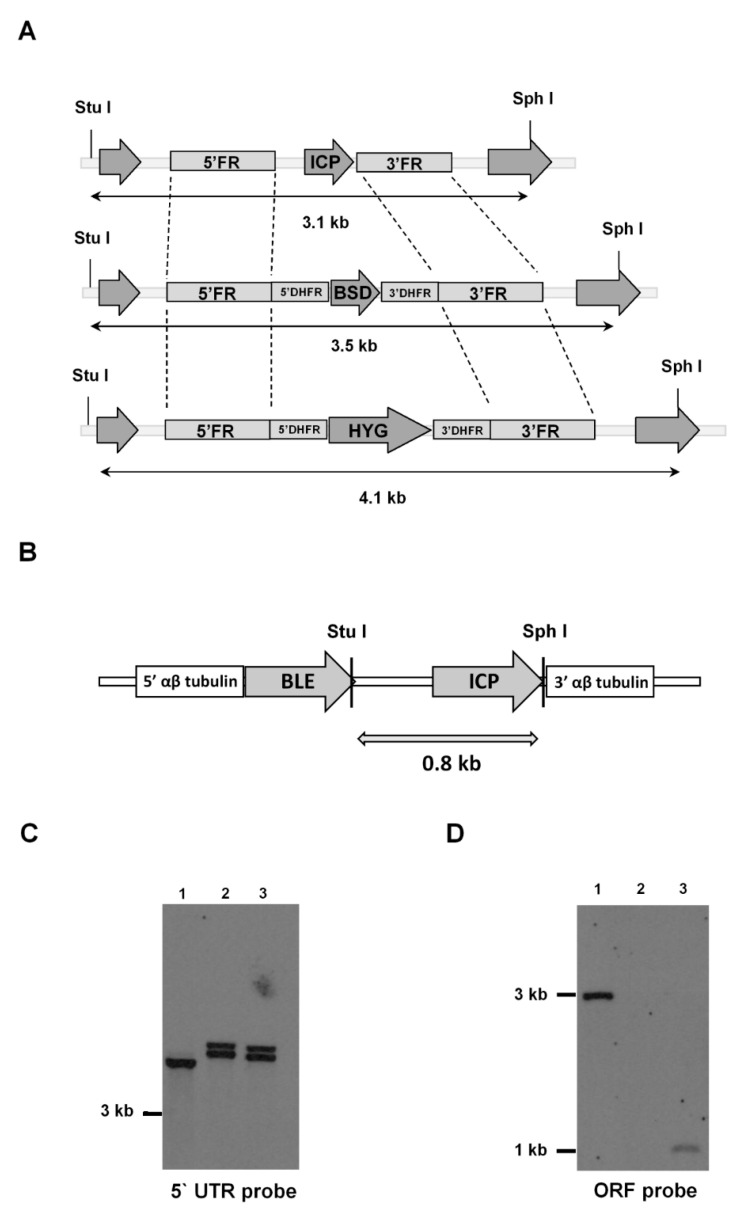
Generation of *T. b. rhodesiense* ICP mutants. (**A**) Schematic representation of the *ICP* locus in wild-type (WT) *T. b. rhodesiense* IL1852 and the constructs containing the same 5′ and 3′ flanking regions (FR), and either a blasticidin (*BSD*) or hygromycin (*HYG*) resistance cassette, for the homologous recombination to generate an *ICP* knock-out line (Δ*icp*). The oligonucleotides used to generate the FR for construction of recombination cassettes are listed in Table 1 (methods). Open reading frames (ORFs) of *ICP* and of flanking genes are shown as arrows. DHFR, dihydrofolate reductase. The predicted DNA fragment sizes after digestion with *Stu*I and *Sph*I are shown. (**B**) Schematic representation of the construct used, containing the phleomycin (*BLE*) resistance cassette, for the re-integration of *ICP* in the tubulin locus of the Δ*icp* line to generate an *ICP* re-expressing line (Δ*icp:ICP*). The predicted DNA fragment size after digestion with *Stu*I and *Sph*I is shown. (**C**,**D**) Genomic DNA isolated from the *T. b. rhodesiense* WT line, a Δ*icp* clone, and a Δ*icp:ICP* clone was digested with *Stu*I and *Sph*I, separated on a 0.8% agarose gel, transferred to a nylon membrane and probed with alkaline phosphatase labeled 5′ FR of *ICP* (**C**) or *ICP* ORF (**D**). Lane 1, WT; Lane 2, Δ*icp*; Lane 3, Δ*icp*:*ICP*.

**Figure 2 ijms-24-00656-f002:**
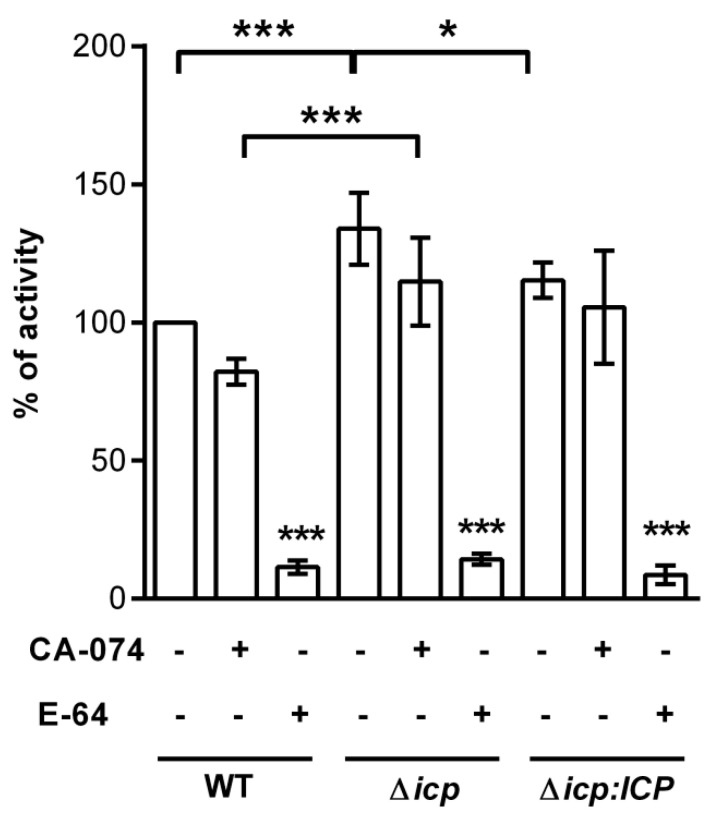
*T. b. rhodesiense* lacking ICP have increased papain-like cysteine peptidase activity. Lysates from 10^6^ bloodstream form (BSF) parasites of *T. b. rhodesiense* WT, Δ*icp*, and Δ*icp*:*ICP* were tested for peptidase activity in a continuous assay using Z-Phe-Arg-MCA as a substrate. Where indicated, 10 μM of CA-074 or 10 μM of E-64 were added to the samples diluted in assay medium for 10 min prior to the addition of the substrate. Experiments were performed in triplicate, two independent times, and are shown as the mean ± SD. Statistical significance between the controls (lysates only) of Δ*icp* and Δ*icp:ICP* at *p* < 0.05 (*) and Δ*icp* and WT at *p* < 0.001 (***), between the E-64 treatments of Δ*icp* and WT at *p* < 0.001 (***), and for each E-64 treatment in relation to the respective (lysate only) control, as assessed by one-way ANOVA with Bonferroni’s post hoc test.

**Figure 3 ijms-24-00656-f003:**
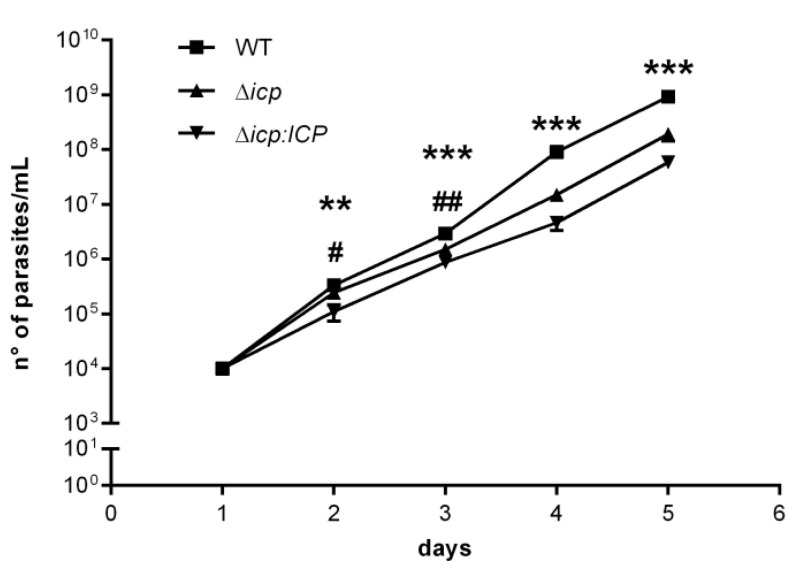
*T. b. rhodesiense* Δ*icp* have altered growth in vitro. Growth curves of BSF *T. b. rhodesiense* WT, Δ*icp*, and Δ*icp:ICP* determined by cell counts on a hemocytometer after cells were seeded in culture medium at 1 × 10^4^ cells/mL. Cultures were diluted to 1 × 10^4^ cells/mL after 3 days and 5 days and the cumulative cell number was determined using the dilution factor. Data are shown as the mean ±SD. Experiment was performed in triplicate and the graph is representative of three independent experiments. Statistical analysis was performed using two-way ANOVA with Tukey’s post hoc test. * indicates significant differences of WT compared to Δ*icp* and Δ*icp:ICP*; and # indicates significant differences between Δ*icp* and Δ*icp:ICP*; where *p* < 0.05 (#), *p* < 0.01 (** and ##), and *p* < 0.001 (***).

**Figure 4 ijms-24-00656-f004:**
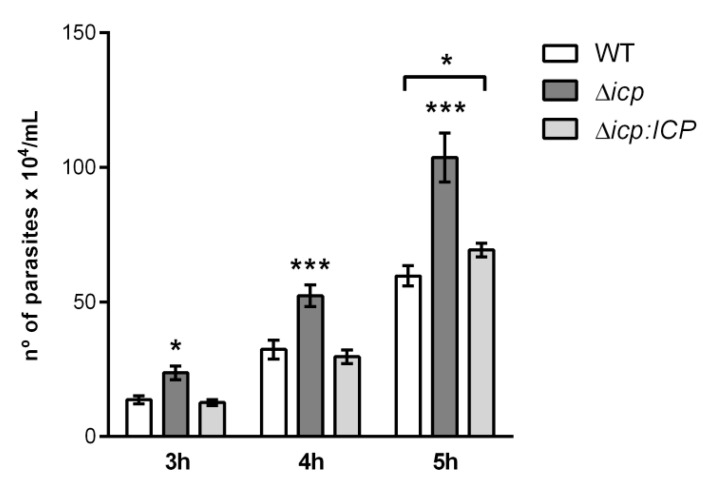
*T. b. rhodesiense* Δ*icp* transmigrate an in vitro blood–brain barrier model more efficiently than wild type. Human brain microvascular endothelial cells (HBMECs) were seeded (5 × 10^3^) on top of Transwell inserts with 3 μm pores and cultured for 5 days, after which BSF parasites (10^6^) of *T. b. rhodesiense* WT, Δ*icp*, and Δ*icp:ICP* were added on top of the HBMEC-containing inserts. The number of parasites present in the bottom chamber was determined by counting aliquots on a Neubauer chamber after 3, 4, and 5 h of incubation. The experiments were performed in triplicate, three independent times, and data are shown as the mean ± SD of a representative experiment. Asterisks indicate statistical significance at *p* < 0.05 (*) and *p* < 0.001 (***) as assessed by two-way ANOVA with Tukey’s post hoc test.

**Figure 5 ijms-24-00656-f005:**
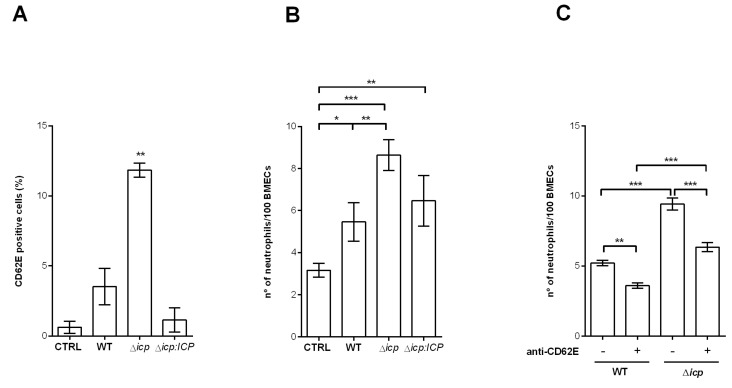
*T. b. rhodesiense* Δ*icp* make HBMECs more susceptible to neutrophil adhesion. (**A**) Human BMECs were cultivated for 3 days, washed, and then incubated with BSF parasites of *T. b. rhodesiense* WT, Δ*icp*, and Δ*icp:ICP* for 18 h. Monolayers were washed, cells were detached, and cell surface E-selectin (CD62E) was detected by flow cytometry. (**B**) HBMECs (5 × 10^4^) were seeded on glass coverslips and cultured for 3 days, after which BSF parasites of *T. b. rhodesiense* WT, Δ*icp*, and Δ*icp:ICP* were added at a 2:1 parasite:cell ratio for 18 h. Cells were then washed and 10^6^ human neutrophils were co-cultivated for 1 h. CTRL refers to control HBMECs not exposed to parasites, and co-cultivated with the neutrophils only. (**C**) Following cultivation on coverslips, HBMECs were exposed to parasites, washed, incubated with anti-E-selectin antibody for 1 h and washed, prior to the addition of neutrophils, as above. After incubation with the neutrophils, cells were washed, fixed with 70% methanol and then Giemsa stained. The number of adhered neutrophils was determined by the counting of 100 HBMECs per coverslip under a light microscope. The experiments were performed in triplicate, (**B**) five independent times or (**C**) two independent times, and data are shown as the mean ± SD of a representative experiment. In (**A**), asterisks indicate significance to all other points. Asterisks indicate statistical significance at *p* < 0.05 (*), *p* < 0.01 (**), and *p* < 0.001 (***) as assessed by one-way ANOVA with Tukey’s post hoc test.

**Figure 6 ijms-24-00656-f006:**
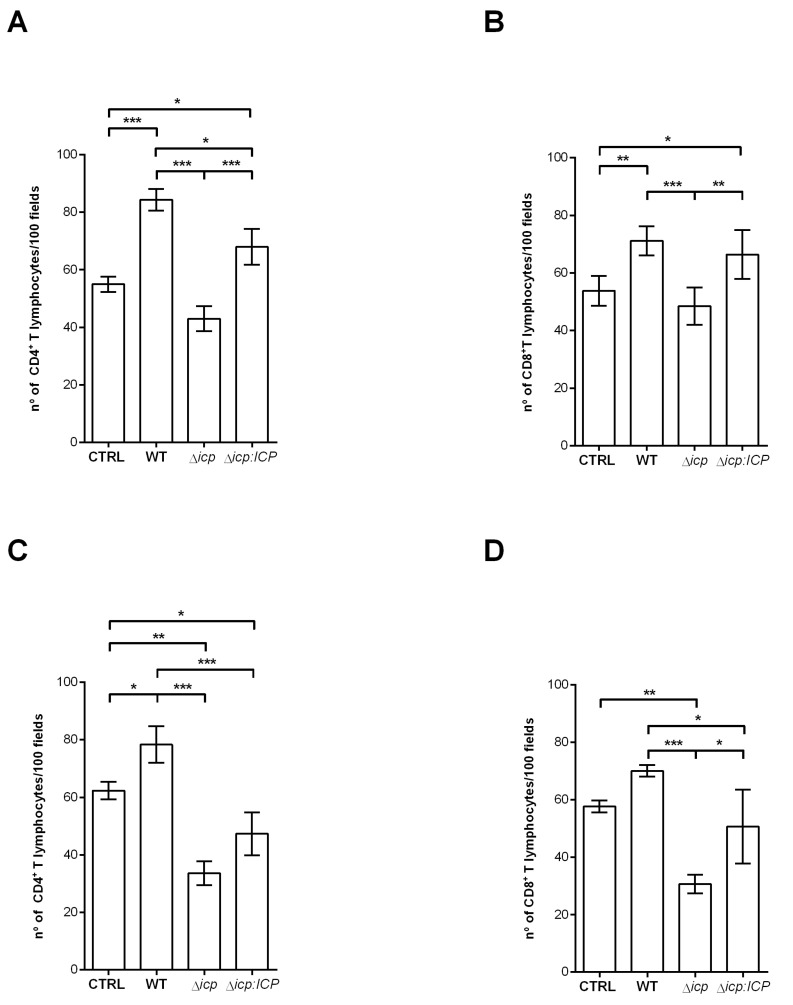
*T. b. rhodesiense* Δ*icp* make HBMECs less susceptible to adhesion by T lymphocytes. Human BMECs (5 × 10^4^) were seeded on glass coverslips and cultured for 3 days, after which BSF parasites of *T. b. rhodesiense* WT, Δ*icp*, and Δ*icp:ICP* were added at a 2:1 parasite:cell ratio for 18 h. CD4^+^ and CD8^+^ T cells (4 × 10^5^) were bead-purified from human blood, and were either (**A**,**B**) non-activated or (**C**,**D**) activated in vitro with 5 µg/mL phytohemagglutinin prior to addition to HBMECs. The cells were then co-cultivated for 2 h. CTRL refers to HBMECs cultivated with either activated or non-activated T cells only. Following this, cells were washed, fixed with 70% methanol and then Giemsa stained. The number of adhered T lymphocytes were determined by the counting of 100 HBMECs per coverslip under a light microscope. The experiments were performed in triplicate, three independent times, and data are shown as the mean ± SD of a representative experiment. Asterisks indicate statistical significance at *p* < 0.05 (*), *p* < 0.01 (**), and *p* < 0.001 (***) as assessed by one-way ANOVA with Tukey’s post hoc test.

**Figure 7 ijms-24-00656-f007:**
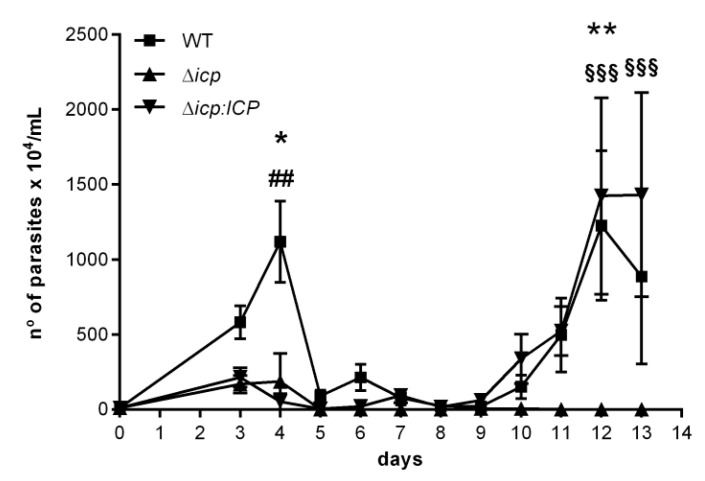
*T. b. rhodesiense* Δ*icp* have reduced virulence in C57BL/6 mice. C57BL/6 mice were infected i.p. with BSF parasites of *T. b. rhodesiense* WT, Δ*icp*, and Δ*icp:ICP* (1 × 10^5^ parasites per animal) and blood parasitemia was determined from days 3 to 13 of infection (starting with n = 5 per group). The graph shows the mean ±SEM of one experiment and is representative of two independent experiments. No mice died over the experimental period. One-way ANOVA with Tukey’s post hoc test was used to determine significant difference between all three groups; where * indicates a significant difference between WT and Δ*icp*, # between WT and Δ*icp:ICP*, and § between Δ*icp* and Δ*icp:ICP* at *p* < 0.05 (*), *p* < 0.01 (** and ##), and *p* < 0.001 (§§§).

**Figure 8 ijms-24-00656-f008:**
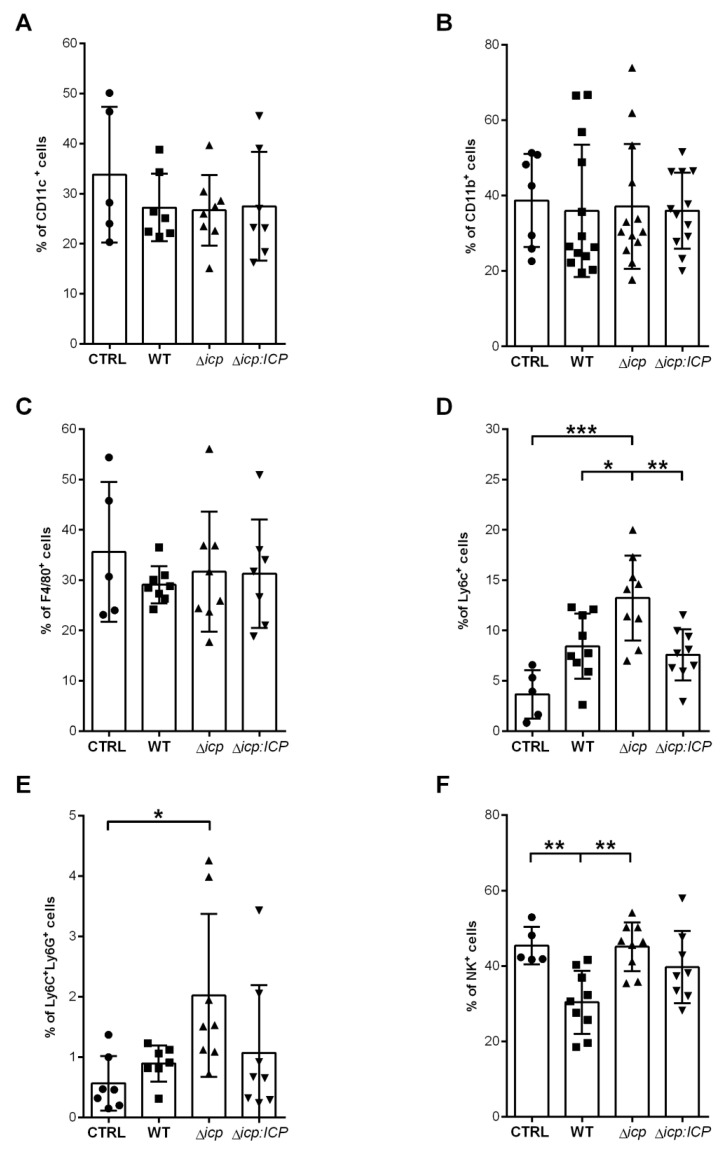
*T. b. rhodesiense* Δ*icp* induces increased recruitment of monocytes to the inoculation site during early infection of C57BL/6 mice. C57BL/6 mice were infected with BSF parasites of *T. b. rhodesiense* WT, Δ*icp*, and Δ*icp*:*ICP* (1 × 10^5^ parasites per animal). On the second day post-infection, mice were euthanized, and the peritoneal cavity was washed with RPMI in order to collect the peritoneal cells. The peritoneal cells were submitted to flow cytometry to assess expression of (**A**) CD11c, (**B**) CD11b, and (**C**) F4/80 within the total cell population, and (**D**) Ly6C^+^Ly6G^−^ cells and (**E**) Ly6C^+^Ly6G^+^ within the CD11b^+^ population. (**F**) Expression of NK1.1 within the total cell population. CTRL represents the non-infected control that was injected with RPMI medium. Graphs show individual points representing each mouse sample and the mean ± SD of two or three combined experiments (n ≥ 3 per group per experiment). Asterisks indicate statistical significance at *p* < 0.05 (*), *p* < 0.01 (**), and *p* < 0.001 (***) as assessed by one-way ANOVA with Tukey’s post hoc test.

**Figure 9 ijms-24-00656-f009:**
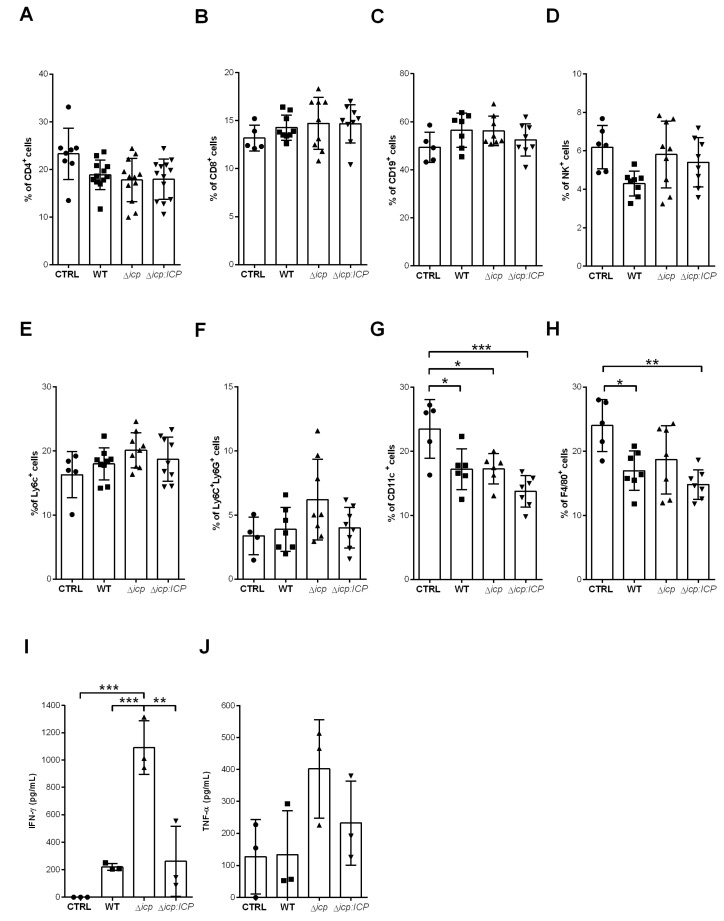
*T. b. rhodesiense* Δ*icp* induces higher inflammatory response in the spleen during early infection of C57BL/6 mice. C57BL/6 mice were infected with BSF parasites of *T. b. rhodesiense* WT, Δ*icp*, and Δ*icp*:*ICP* (1 × 10^5^ parasites per animal). On the second day post-infection, mice were euthanized, spleens were removed and macerated, then the splenocytes were submitted to flow cytometry or were cultured for 24 h, after which the supernatant was collected and cytokine levels determined by ELISA. The splenocytes were submitted to flow cytometry to assess expression of (**A**) CD4, (**B**) CD8, (**C**) CD19, and (**D**) NK1.1 within the total cell population. The proportion of the (**E**) Ly6C^+^Ly6G^−^ and (**F**,**L**) Ly6C^+^Ly6G^+^ populations within the CD11b^+^ was determined, while the expression of (**G**) CD11c and (**H**) F4/80 within the total cell population was assessed. Graphs show individual points representing each mouse sample and the mean ± SD of two independent experiments combined (n ≥ 3 per group per experiment). The splenocyte culture supernatants were tested for (**I**) IFN-γ and (**J**) TNF-α. CTRL represents the non-infected control that was injected with RPMI medium. Graphs show individual points representing each mouse sample and the mean ± SD of one experiment (n = 3 or 4 per group per experiment) and are representative of the profile of two independent experiments. Asterisks indicate statistical significance at *p* < 0.05 (*), *p* < 0.01 (**), and *p* < 0.001 (***) as assessed by one-way ANOVA with Tukey’s post hoc test.

**Figure 10 ijms-24-00656-f010:**
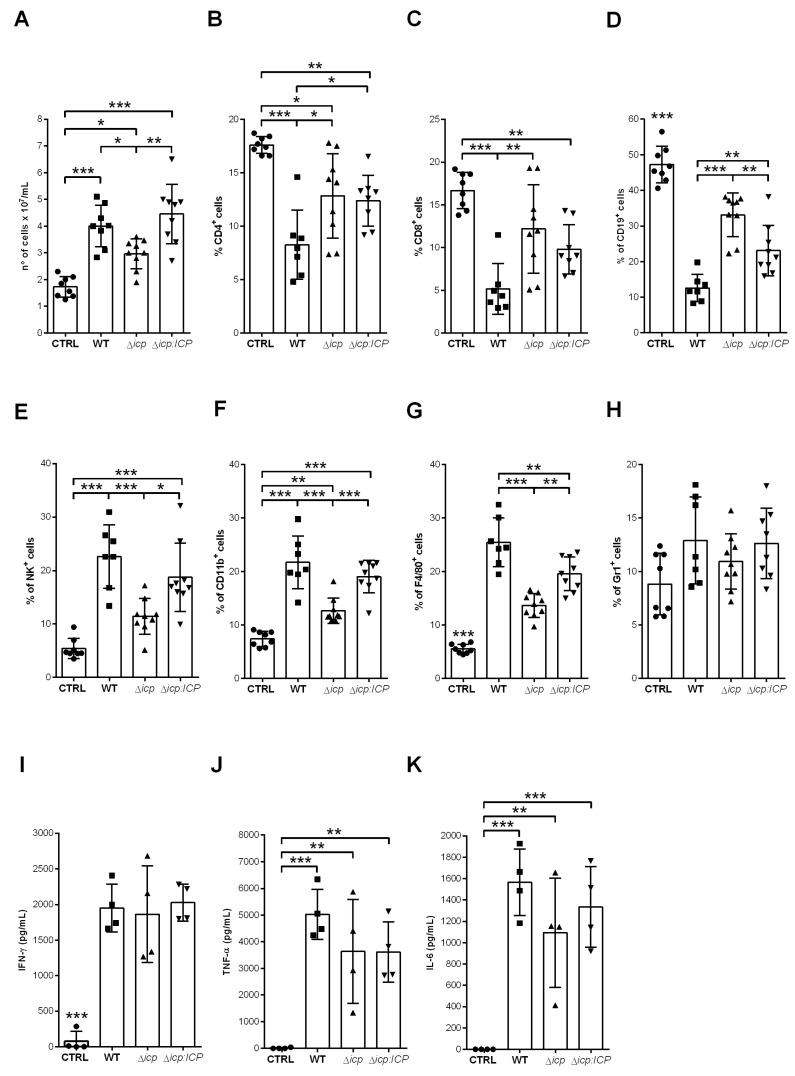
C57BL/6 infected with *T. b. rhodesiense* lacking ICP dampen the immune response in the spleen by day 14. C57BL/6 mice were infected with BSF parasites of *T. b. rhodesiense* WT, Δ*icp*, and Δ*icp*:*ICP* (1 × 10^5^ parasites per animal). After 14 days, mice were euthanized and the spleens were collected, then macerated in order for (**A**) the cellularity to be determined. The splenocytes were submitted to flow cytometry (**B**–**H**) or were cultured for 24 h, after which the supernatant was collected. The splenocytes were submitted to flow cytometry to assess expression of (**B**) CD4, (**C**) CD8, (**D**) CD19, (**E**) NK1.1, (**F**) CD11b, (**G**) F4/80, and (**H**) Gr1 within the total cell population. Graphs show individual points representing each mouse sample and the mean ± SD of two independent experiments combined (n ≥ 4 per group per experiment). The splenocyte culture supernatants were tested by ELISA to determine the levels of (**I**) IFN-γ, (**J**) TNF-α, and (**K**) IL-6. CTRL represents the non-infected control. Graphs show individual points representing each mouse sample and the mean ± SD of one experiment (n = 4 per group) and are representative of the profile of two independent experiments. Statistically significant differences were determined by one-way ANOVA with Tukey’s post hoc test, where *p* < 0.05 (*), *p* < 0.01 (**), and *p* < 0.001 (***).

**Table 1 ijms-24-00656-t001:** Oligonucleotides used to generate constructs.

Oligo	Sequence
OL1609 5′ FW	CGGCGGCCGCGGTGGAGATTAAAAAAAGAAAAAAGTG
OL1610 5′ REV	CGTCTAGAGCAACAAAAATCAATGACATG
OL1611 3′ FW	CGGGGCCCGGTATGTGGAAGTGGAGAAG
OL1612 3′ REV	CGGGGCCCGATATCGGCGGGATGGAGTAAACATA
NT90 ORF FW	GGCTC-GAGCTATGCGGTGGCCTCGACGTGAATG
NT91 ORF REV	GGC-GCATATGTCCCACAACCTATTTACTGAGG

Underlined, restriction site.

## Data Availability

All the data generated is available in the manuscript or upon request to the authors.

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
