# Peer review of "Trypanosoma brucei rhodesiense Inhibitor of Cysteine Peptidase (ICP) Is Required for Virulence in Mice and to Attenuate the Inflammatory Response"

_ijms, 2022, doi:10.3390/ijms24010656_

Round 1
Reviewer 1 Report
The research article entitled “Trypanosoma Brucei Rhodesiense Inhibitor of Cysteine Peptidase (ICP) Is Required for Virulence in Mice and to Attenuate the Inflammatory Response” encompasses an interesting study where the authors have tested the role of an endogenous cysteine peptidase inhibitor (ICP) in virulence of parasite. The study provides strong evidence for the intricate role of ICP in host parasite interaction and parasite survival. The paper is well written, the experiments are appropriately designed and well executed. The authors also provide a very elaborate and sensible interpretation of the results. I believe the article is suitable for publication in IJMS with minor revisions.
I would suggest the authors to rewrite the result section of cysteine peptidase activity assay. The experiment doesn’t provide enough evidence to connect the observed reduction in peptidase activity to TbCATL inhibition specifically, considering the substrate used in the study is not TbCATL/B specific. Also, possible involvement of other ICP targets contributing to the peptidase activity can’t be ruled out completely. In general, this particular experiment is helpful only in establishing the increased overall cysteine peptidase activity in Ñicp parasite lysate.
Possibility of ICP targets other than TbCATL contributing to at least some of the response elicited by ICP deficient parasites should also be considered while interpreting the data from other assays and animal model studies. I would like to see the author’s thoughts about ICP specificity in discussion.
Author Response
I would suggest the authors to rewrite the result section of cysteine peptidase activity assay. The experiment doesn’t provide enough evidence to connect the observed reduction in peptidase activity to TbCATL inhibition specifically, considering the substrate used in the study is not TbCATL/B specific. Also, possible involvement of other ICP targets contributing to the peptidase activity can’t be ruled out completely. In general, this particular experiment is helpful only in establishing the increased overall cysteine peptidase activity in Ñicp parasite lysate.
Re: We thank the reviewer for pointing this out. We have amended the results section to mention C1A peptidase activity, according to E64 inhibition, and removed considerations on TbCATL versus TbCATB (lines 108-116).
Possibility of ICP targets other than TbCATL contributing to at least some of the response elicited by ICP deficient parasites should also be considered while interpreting the data from other assays and animal model studies.
Re: We have modified the discussion to include considerations about additional ICP targets.
I would like to see the author’s thoughts about ICP specificity in discussion.
Re: We have modified the discussion to comment this point (lines 378-384).
Reviewer 2 Report
The article 'Trypanosoma Brucei Rhodesiense Inhibitor of Cysteine Peptidase (ICP) Is Required for Virulence in Mice and to Attenuate the Inflammatory Response' by Costa et al. investigated the role of inhibitor of cysteine peptidases (ICP) in T. b rhodesiense bloodstream form and propose that ICP helps to downregulate inflammatory responses that contribute to the control of infection. The results presented here are informative and relevant to the readership of 'IJMS.' The data presented generally supports the conclusion. The authors need to address some critical concerns in the current version of the manuscript before its publication. The issues are listed below.
The Abstract is written in very casual language. It needs to be rewritten.
How do the authors explain the failed complementation in Δicp:ICP?
What happens to the increased recruitment of monocytes in Δicp during early infection? Do they not differentiate into macrophages?
Why did the authors decide to evaluate the immune response at day 2 post-infection when the first wave of parasitemia was observed around day 4?
How do the authors explain reduced growth but increased immune response at day 2 in Δicp?
Have the authors looked at the other inflammatory markers such as IL-4, IL-10, IL-12, or TGF-β as these cytokines mediate immune response during the parasite infection?
There are many verb tense inconsistencies throughout the manuscript. Further, the writing style, typographical and grammatical errors should be corrected in the revised version of the manuscript.
Author Response
The Abstract is written in very casual language. It needs to be rewritten.
Re: The abstract was modified.
How do the authors explain the failed complementation in Δicp:ICP?
Re: Δicp:ICP complemented nearly all the phenotypes displayed by Δicp to wild type levels, i.e., traversal of the BBB model, induction of endothelial markers, changes in leukocyte adhesion, cellular recruitment in vivo and cytokine profiles. Δicp:ICP failed to complement slower in vitro growth of BSF, and lower parasite burden in the first wave of parasitemia, which is compatible with the observation of slower growth in vitro. However Δicp:ICP also behaved similarly to wild type parasites, at the second wave of parasitemia. As discussed in the manuscript, it is possible that the complementation of the growth phenotype requires expression of ICP to nearly identical levels found in wild-type, which is challenging to achieve in complemented lines. Another possibility is that ICP affects the differentiation from long-slender to stumpy forms, and if the parasites remain as non-multiplying stumpies for longer, for example, this could result in lower culture densities in vitro. In this scenario, it is possible that the levels of ICP re-expression in the complemented line were insufficient to adequately inhibit all excess of peptidase activity.
What happens to the increased recruitment of monocytes in Δicp during early infection? Do they not differentiate into macrophages?
Re: The fate of monocytes recruited to tissues is a timely event that varies depending on the cytokine combination in the microenvironment. While some may differentiate to macrophages or to dendritic cells, others may remain as immature monocytes. At day 2 post-infection, it is expected that continuous waves of cellular recruitment are still ongoing, which would maintain the population of undifferentiated recruited monocytes.
Why did the authors decide to evaluate the immune response at day 2 post-infection when the first wave of parasitemia was observed around day 4?
Re: As pointed in line 262, we aimed to address how the innate response was mounting in the initial stages after infection. Initial immune responses such as DC maturation, innate cytokine secretion and cellular recruitment may start to occur as early as 18hrs after a given stimulus. We wanted to determine if there were changes in how the innate responses were building, which could have consequences for the control of parasite burden a few days later.
How do the authors explain reduced growth but increased immune response at day 2 in Δicp?
Re: It is known that mononuclear cells, mainly macrophages, are the main controllers of T. brucei BSF during mice infections via parasite phagocytosis. It is possible that the increased number of monocytes and/or increased activation of macrophages contributed to reduce parasite numbers in the blood during the first wave. This is mentioned in the discussion (lines 459-462).
Have the authors looked at the other inflammatory markers such as IL-4, IL-10, IL-12, or TGF-β as these cytokines mediate immune response during the parasite infection?
Re: We have analysed IL10 and IL12, and no differences were observed among experimental groups. This information was added to the discussion.
There are many verb tense inconsistencies throughout the manuscript. Further, the writing style, typographical and grammatical errors should be corrected in the revised version of the manuscript.
Re: We thank the reviewer for pointing this out. The ms was reviewed throughout.
Round 2
Reviewer 2 Report
The revised version of this manuscript has addressed all the issues raised in the previous version of this paper. Hence, I endorse the publication.